# The Role of MicroRNAs in Aortic Stenosis—Lessons from Recent Clinical Research Studies

**DOI:** 10.3390/ijms241713095

**Published:** 2023-08-23

**Authors:** Anna Krauze, Grzegorz Procyk, Aleksandra Gąsecka, Izabela Garstka-Pacak, Małgorzata Wrzosek

**Affiliations:** 1Department of Biochemistry and Pharmacogenomics, Medical University of Warsaw, Banacha 1, 02-097 Warsaw, Poland; krauze-ania@wp.pl (A.K.); garstkaiza@gmail.com (I.G.-P.); 21st Chair and Department of Cardiology, Medical University of Warsaw, Banacha 1A, 02-097 Warsaw, Poland; aleksandra.gasecka@wum.edu.pl; 3Doctoral School, Medical University of Warsaw, 02-091 Warsaw, Poland

**Keywords:** aortic stenosis, calcific aortic valve disease, microRNAs, biomarkers, aortic valve replacement

## Abstract

Aortic stenosis (AS) is the most prevalent primary valve lesion demanding intervention. Two main treatment options are surgical aortic valve replacement or transcatheter aortic valve implantation. There is an unmet need for biomarkers that could predict treatment outcomes and become a helpful tool in guiding Heart Team in the decision-making process. Micro-ribonucleic acids (microRNAs/miRs) have emerged as potential biomarkers thoroughly studied in recent years. In this review, we aimed to summarize the current knowledge about the role of miRNAs in AS based on human subject research. Much research investigating miRNAs’ role in AS has been conducted so far. We included 32 original human subject research relevant to the discussed field. Most of the presented miRNAs were studied only by a single research group. Nevertheless, several miRNAs appeared more than once, sometimes with high consistency between different studies but sometimes with apparent discrepancies. The molecular aspects of diseases are doubtlessly exciting and provide invaluable insights into the pathophysiology. Nevertheless, translating these findings, regarding biomarkers such as miRNAs, into clinical practice requires much effort, time, and further research with a focus on validating existing evidence.

## 1. Introduction

Aortic stenosis (AS) is the most prevalent primary valve lesion demanding intervention–either surgical or transcatheter [1]. The main symptoms are chest pain, palpitations, dizziness, and dyspnea. Nevertheless, AS remains symptomless for a long time. It leads to left ventricular hypertrophy (LVH), impairing the function of the left ventricle. The assessment of AS severity differs depending on the type of AS: (i) high-gradient AS; (ii) low-flow, low-gradient AS with reduced ejection fraction (EF); (iii) low-flow, low-gradient AS with preserved EF; (iv) normal-flow, low-gradient AS with preserved EF. In the case of symptomatic patients with severe AS, early intervention is highly recommended [1].

Two main treatment options are surgical aortic valve replacement (AVR) or transcatheter aortic valve implantation (TAVI). The decision-making process is complex and should be carried out after the Heart Team evaluation [2]. The main factors favoring TAVI are older age, high surgical risk, and previous cardiac surgery—conversely, younger age and lower surgical risk advantage the choice of surgical AVR [1]. Nevertheless, there is a significant group of patients in a ‘grey zone’ when it is not clear what option will benefit the patient more. There is an unmet need for biomarkers that could predict treatment outcomes and become a helpful tool in guiding the Heart Team in the decision-making process. 2021 ESC/EACTS Guidelines for the management of valvular heart disease note it as a gap in evidence (the lack of tools for risk stratification): (i) to decide if intervention is required; (ii) to choose the best treatment option. Another gap in the evidence regarding AS is the pathophysiology of progression. A better understanding of this process could lead to novel therapeutics. Both gaps in evidence could potentially be filled by the micro-ribonucleic acids (miRNAs/miRs)—promising biomarkers, which have been thoroughly studied in recent years.

MiRNAs are a subtype of ribonucleic acids (RNAs) consisting of about 22 nucleotides [3]. They are synthesized in the nucleus and then further processed in the cytoplasm into their final form, in which they can stay inside the cell or be released into circulation and other fluids [4]. Therefore, miRNAs can be assessed in tissues, the blood (circulating miRNAs), or even in cerebrospinal fluid [5]. MiRNAs play a crucial role in regulating gene expression [6]. They bind to mRNAs, which are repressed or degraded based on the degree of complementarity between miRNA and mRNA. MiRNA expression is altered in many diseases [7,8]. Therefore, not only can they become diagnostic, prognostic, and predictive biomarkers, they are also potential targets for novel therapeutic agents [9].

The pathophysiology of AS development includes inflammation, endothelial dysfunction, and osteogenic differentiation of interstitial cells. All these lead to progressive calcification and subsequent outflow obstruction [10]. Since miRNAs are critical regulators of gene expression, they play an essential role in processes like inflammation or osteogenic differentiation [11]. The dysregulation of miRNA expression leads to the dysregulation of gene expression they control. The involvement of miRNA in AS has been the subject of research and is discussed in detail in a further part of this review.

In this review, we aimed to summarize the current knowledge about the role of miRNAs in AS based on human subject research.

## 2. MicroRNAs in Patients Suffering from Aortic Stenosis

In our review, we included only original human subject research. Reviews, letters to the editors, and commentaries were not included. We searched the PubMed Database, and after a thorough evaluation of its records, we included 32 original human subject studies relevant to the discussed field.

We divided these studies into the following parts: (i) altered microRNA expression in patients with aortic stenosis, (ii) altered microRNA expression in patients with calcific aortic valve disease, (iii) differences in microRNA expression between aortic stenosis patients with bicuspid and tricuspid aortic valves, (iv) comparison of microRNA expression between calcified and adjacent non-calcified aortic valve tissues in the same patients, (v) predictive values of microRNAs after aortic valve replacement and transcatheter aortic valve implantation (Figure 1).

### 2.1. Altered MicroRNA Expression in Patients with Aortic Stenosis

All studies discussed further in this subsection with additional data are summarized in Table 1. First, Nigam et al. investigated patients undergoing AVR due to a bicuspid aortic valve (BAV). They compared miRNA levels between patients with AS and aortic insufficiency (AI). The first group was found to have decreased levels of miR-26a and miR-195 compared to the latter [12]. Villar et al. measured the expression of miR-21 in myocardial samples from AS patients undergoing AVR and compared the results to the surgical controls. They found that the levels of miR-21 were significantly higher in patients suffering from AS. They also compared plasma levels of miR-21 between this group of patients with AS and healthy volunteers. Similarly, patients with AS exhibited higher levels of this biomarker [13].

Beaumont et al. analyzed the role of miRs in the progression of myocardial fibrosis in patients with AS. The authors included patients suffering from AS who underwent AVR and divided them according to their myocardial collagen volume fraction (CVF)—a marker of fibrosis. It was found that patients with severe fibrosis (SF) presented lower myocardial expression levels of miR-18b and miR-122 compared to patients without SF. Moreover, the authors found an inverse correlation of miR-122 with CVF and transforming growth factor-β type 1 (TGF-β1) [14].

Røsjø et al. examined the levels of miR-210 in patients with moderate to severe AS and compared them to healthy controls. The levels of miR-210 were increased in patients with AS compared to controls. Moreover, the authors showed that higher levels of miR-210 were significantly associated with increased mortality in patients with AS during a median follow-up of about 3.5 years [15]. Coffey et al. measured the levels of several miRs in patients with AS, comparing them to control individuals. The authors subdivided people from both groups into participants with and without coexisting coronary artery disease (CAD). Within the population with CAD, it was found that patients with AS had upregulated miR-22-3p, miR-24-3p and downregulated miR-382-3p compared to controls. Meanwhile, in the population without coexisting CAD, patients with AS had upregulated miR-21-5p and downregulated miR-22-3p compared to controls. The authors concluded that the mixed picture of miR expression they obtained highlighted that implementing miRs as biomarkers might be complicated and require further refinement [16]. In another study by Coffey et al., miRNA expression in aortic valves obtained from (i) patients with AS undergoing AVR and (ii) non-diseased cadavers was analyzed. They found that miR-21-5p and miR-221-3p were upregulated, while miR-30e-5p, miR-122-5p, and miR-625-5p were downregulated in patients with AS compared to controls [17].

Interesting research was conducted by Gallego et al. They recruited patients with AS undergoing AVR and divided them based on the cardiomyocyte apoptotic index (CMAI) into two groups: (i) with low CMAI and (ii) with high CMAI. Moreover, cardiovascular-disease-free control subjects constituted the additional control group. It was found that miR-10b, miR-125b-2*, and miR-338-3p were downregulated in both groups of patients with AS compared to controls. Moreover, all three miRs were downregulated in patients with high CMAI compared to patients with low CMAI. Interestingly, all the levels of these miRs correlated inversely with CMAI [18].

Takahashi et al. aimed to study the expression of several miRs in circulating osteogenic progenitor cells (COPCs) from patients with calcific aortic valve disease (CAVD) undergoing AVR or TAVI. The authors showed upregulation of miR-30c and downregulation of miR-31, miR-106a, miR-148a, miR-204, miR-211, and miR-424 in patients with AS compared to control individuals. Moreover, it was found that miR-30c correlated weekly with the degree of aortic valve calcification. Interestingly, the levels of all investigated miRs changed after the procedure (AVR or TAVI): the level of miR-30c decreased, while the levels of remaining miRs increased. AVR caused more significant changes in miR levels than TAVI [19].

Beaumont et al. published research conducted on the same population as the previously mentioned research by Gallego et al., but this time they focused on different miRs. Moreover, in this investigation, they measured miR expression in both the myocardium and the blood. They found that the miR-19b and miR-133a level was lower in patients with AS than in control subjects in the myocardium and in the blood. Moreover, they found that miR-19b was downregulated in patients suffering from AS with coexisting heart failure (HF) compared to patients with AS but without HF [20].

Differently from previously described research, Song et al. studied the expression of miRs in aortic valve interstitial cells (AVICs) in patients with AS undergoing AVR. Patients undergoing heart transplantation due to cardiomyopathy served as controls. The authors found that miR-204 was downregulated and miR-486 upregulated in patients with AS compared to controls [21]. Xu et al. compared miR levels in aortic valve samples between patients with AS and patients with AI. The levels of miR-26a, miR-374b, and miR-939 appeared lower, and miR-214 was higher in patients with AS compared to patients with AI [22].

Similarly, Fiedler et al. assessed several microRNAs expression in the aortic valves of patients suffering from AS undergoing AVR. They compared it to the expression of these microRNAs in the aortic valves of healthy controls from a biobank. They showed increased miR-21, miR-24, and miR-143 levels in patients with AS. Further experiments proved that miR-143 regulated the expression of matrix gla protein (MGP) via direct binding to its 3′UTR region. Noteworthily, MGP plays a crucial role in maintaining homeostasis in the aortic valves [23].

Interestingly, Petrkova et al. investigated the relationship between atherosclerosis in patients with AS and miRNA expression. They included patients undergoing AVR due to AS. They showed higher levels of miR-146a in valvular tissues obtained from patients with atherosclerosis compared to non-atherosclerotic patients [24].

To summarize, many studies have revealed that patients suffering from AS presented with changed expression of various miRs. This applied to measurements conducted in plasma, aortic valve tissues, myocardium, and even COPCs. Although the results are undoubtedly promising, the multitude of studied miRs hampers drawing firm cause-and-effect conclusions.

### 2.2. Altered MicroRNA Expression in Patients with Calcific Aortic Valve Disease

As in the previous paragraph, all studies discussed in this subsection with additional data are summarized in Table 2. Song et al. compared the expression of miR-204 in the aortic valves of patients with CAVD undergoing AVR to its expression in patients with cardiomyopathy undergoing heart transplantation. They revealed the downregulation of miR-204 in CAVD patients. The authors suggested the potential role of exogenous miR-204 in inhibiting the valvular calcification [25]. Quite similarly, Zheng et al. investigated miR-214 in CAVD patients undergoing AVR and compared them to patients undergoing AVR due to AI. They proved that miR-214 was upregulated in CAVD patients compared to the latter group. It applied to the measurements both in the blood and in the aortic valve samples. The authors concluded that miR-214 might facilitate the inflammatory reaction and thus promote calcification [26].

Jiang et al. further studied the importance of miRNAs in CAVD. They compared miRNA levels between calcified and non-calcified aortic valves. They showed downregulation of miR-135-5p, miR-204-5p, miR-335-3p, and miR-664a-3p in valves from CAVD patients compared to valves from patients undergoing heart transplantation. They further investigated miR-664a-3p in vitro and discovered its involvement in the regulation of osteogenic differentiation of human aortic valve interstitial cells (hAVICs) [27].

Wang et al. focused on the role of miR-629-3p in AS. They compared its expression levels in aortic valves between patients with CAVD and without CAVD, both undergoing AVR. Most importantly, they observed the downregulation of miR-629-3p in valves from CAVD patients compared to the control group. Further in vitro investigation revealed that miR-629-3p and its target, TAGLN, played an anti-calcification role in the osteogenic differentiation of hAVICs [28]. Likewise, Yan et al. included patients with CAVD undergoing AVR and compared them to patients suffering from aortic regurgitation with non-calcified aortic valves. They discovered a decreased level of miR-138-5p in CAVD patients compared to the latter group. The authors concluded that miR-138-5p could be involved in inhibiting valve calcification [29].

Finally, Yang et al. investigated the involvement of miR-22 in CAVD pathogenesis. They compared miR-22 expression in aortic valve samples between CAVD patients undergoing AVR and non-CAVD patients undergoing heart transplantation. It was found that miR-22 was upregulated in CAVD patients compared to non-CAVD patients. The authors conducted further in vitro assessments, concluding that miR-22 accelerated the osteogenic differentiation of hAVICs [30].

All the discussed studies show that miRNAs may play a role in the pathogenesis of CAVD. Although most studies focused only on altered miRNA expression, which can be only a result, not a cause, insights from in vitro analyses suggest miRNA involvement in the calcification process. The regulation of osteogenic differentiation of hAVICs is one of the most critical processes in which miRNAs are involved in this context.

### 2.3. Difference in MicroRNA Expression between Aortic Stenosis Patients with Bicuspid and Tricuspid Aortic Valve

Consistently, we summarized all studies discussed in this subsection with additional data in Table 3. Du et al. recruited patients with AS undergoing AVR. The authors compared miR expression levels between patients with a BAV and a tricuspid aortic valve (TAV). It was found that the levels of miR-195 and miR-486 were lower in the BAV samples than in the TAV samples [31]. Similarly, Nader et al. aimed to study the expression levels of miRs in bicuspid and tricuspid valve calcification in patients with AS undergoing AVR. They found overexpression of miR-92a in patients with a BAV compared to patients with a TAV. The authors named miR-92 as a potential aortic valve calcification biomarker [32].

Like in the previous research, Zheng et al. conducted a study comparing the levels of various miRNAs in aortic valve samples between patients with a bicuspid aortic valve (BAV) and a tricuspid aortic valve (TAV), both suffering from AS and undergoing AVR. They showed increased levels of miR-330-3p, miR-659, and miR-663 and decreased levels of miR-146 in patients with BAVs compared to control patients with TAVs. They further studied the role of miR-330-3p in vitro and demonstrated that the upregulation of miR-330-3p promoted the calcification progress [33].

Not only does miRNA expression differ between patients with AS and healthy/control individuals, but there is also a difference between AS patients with BAVs and TAVs. Although only a few studies have investigated this problem, it seems intriguing and worth studying in future research.

### 2.4. Comparison of MicroRNA Expression between Calcified and Adjacent Non-Calcified Aortic Valve Tissues in the Same Patients

All studies discussed in this subsection with additional data are summarized in Table 4. Progressive calcification is an essential factor in the deterioration of AS. Zhang et al. included patients suffering from AS who underwent AVR. The authors aimed to compare the expression level of miR-30b between calcified and non-calcified parts of aortic valve leaflets. They showed that miR-30b was downregulated in calcified areas, pointing out that miR-30b could serve as a potential target for novel therapies to limit calcification progression in AS [34]. Likewise, Jiao et al. obtained aortic valve samples from patients with calcific aortic valve disease (CAVD) undergoing AVR. They compared calcific aortic valves containing calcific nodules to adjacent non-calcific aortic valve tissues from the same patient. They proved the upregulation of miR-638 in calcific compared to non-calcific aortic valves. Based on further in vitro research, they suggested that miR-638 was an inhibitor of osteogenic differentiation of hAVICs [35].

Lu et al. conducted similar research to Jiao et al., including patients with degenerative CAVD undergoing AVR. They analyzed miR-138 expression in calcified and non-calcified areas of aortic valve samples obtained from included patients. They demonstrated the significant downregulation of miR-138 in calcific compared to non-calcific aortic valves. In further in vitro studies, they obtained results suggesting that miR-138 plays an inhibitory role in the osteogenic differentiation of hAVICs, which seems counterintuitive, considering the results presented by Jiao et al. [36].

Finally, Yu et al. investigated miR-25-3p in CAVD patients undergoing AVR. Like other research teams, they compared expression levels between diseased aortic valve tissue and adjacent normal tissue. MiR-25-3p appeared to be upregulated in calcific valves. The authors concluded that miR-25-3p and the whole TGFBR2/miR-25-3p/TWIST1 axis played a role in osteoblast differentiation in hAVICs [37].

All the studies above show that calcific and non-calcific parts differ in miRNA expression. This suggests that miRNA may be directly involved in calcification and further aortic valve degeneration. Almost every study observed the involvement of hAVICs and their osteogenic differentiation in the studied process.

### 2.5. Predictive Value of MicroRNAs after Aortic Valve Replacement and Transcatheter Aortic Valve Implantation

Table 5 summarizes all studies discussed in this subsection with additional data. Interestingly, Villar et al. analyzed the impact of various factors on postoperative left ventricular mass regression in patients with AS after AVR. It was revealed that increased myocardial expression of miR-133a was an independent predictor of left ventricular mass reduction one year after AVR [38]. In a paper by García et al., the same research group further explored miR-133a as an important prognostic factor for left ventricular mass reduction after AVR in patients with AS. They investigated the potential predictive utility of this biomarker when measured in plasma. First, they showed that plasma and myocardial levels of miR-133a correlated directly, suggesting that the myocardium is an essential source of plasmatic miR-133a. Consistently, the level of miR-133a in plasma was found to be a positive predictor for the regression of left ventricular hypertrophy one year after AVR [39].

Chen et al. aimed to investigate miR-1, miR-133, and miR-378 in predicting LVH in patients with AS. They found that miR-378 was an independent predictor of this condition. The authors also found that the miR-1, miR-133, and miR-378 levels were significantly lower in patients with AS than in healthy controls [40]. Kleeberger et al. investigated various miRs as potential predictors of cardiac function after TAVI in patients suffering from AS. They found a negative correlation between the level of miR-206 measured one day after TAVI and left ventricular ejection fraction (LVEF). The authors stated that left ventricular function was better in patients with lower miR-206 expression [41].

Fabiani et al. included patients with severe AS and made comparisons within this group based mainly on TTE parameters. They found that (i) patients with reduced ejection fraction had increased levels of miR-1 and miR-133, (ii) patients characterized by low flow condition had increased levels of miR-1 and miR-21, (iii) miR-21 was increased in patients with reduced global longitudinal strain, (iv) during one-year follow-up miR-21 was found to be an independent predictor of reverse remodeling while miR-29 independently predicted systolic function increase [42]. Eyileten et al. included patients diagnosed with severe AS qualified from TAVI. The authors measured miR-125a-5p, miR-125b, and miR-223 levels in platelet-depleted plasma before and after the TAVI procedure. The authors reported that miR-125b and miR-223 levels increased after TAVI. Moreover, the baseline level of miR-223 was found to predict major adverse cardiac and cerebrovascular events after TAVI in a univariate analysis but not in a multivariate analysis [43].

As discussed above, miRNA utility goes beyond diagnosis. It may have potential prognostic and predictive value. There is a long way to go in implementing them in routine clinical practice, but it is undoubtedly exciting and connects science and personalized medicine.

## 3. Conclusions

Much research investigating miRNAs’ role in AS has been conducted so far. Scientists have often assessed miRNAs using microarrays and validated the results with quantitative reverse transcription-polymerase chain reaction (qRT-PCR). In our review, we focused only on the results confirmed via qRT-PCR. After presenting all the existing knowledge in the field, some important and exciting issues were discussed.

Most of the presented miRNAs were studied only by a single research group. Nevertheless, several miRNAs have appeared more than once, sometimes with high consistency between different studies but sometimes with apparent discrepancies.

Within those characterized by high consistency, miR-21 and miR-204 are undoubtedly worth mentioning. MiR-21 was the research subject in five cited studies [13,16,17,23,42]. Each time, it was found to be upregulated in patients with AS. Nevertheless, miR-21 has substantial potential biological function, and thus, it is not easy to speculate if and how it may be involved in AS without further research. Conversely, miR-204 was found to be downregulated in four cited studies [19,21,25,27]. Like miR-21, miR-204 potentially targets numerous genes, making it difficult to proceed with cause-and-effect reasoning. Nevertheless, it is worth mentioning that out of nine referenced studies, only one was published within the past 3 years. This indicates quite a serious limitation, i.e., the lack of more recent analyses.

On the other hand, miR-22 and miR-133 are worth discussing due to some inconsistencies. Yang et al., in their very recent research, discovered that miR-22 is downregulated in the valves of patients with CAVD. Coffey et al. demonstrated partially different but reasonably explainable results a few years earlier: patients with AS without coexisting CAD were found to have decreased levels of miR-22, as in the study by Yang et al. However, patients with AS and coexisting CAD presented increased levels of miR-22 compared to controls. 

The results from the research of Fabiani et al. regarding miR-133a is more surprising [42]. Two research groups found that miR-133 is downregulated in AS [20,40]. Consistently, increased levels of miR-133 were found to be an independent predictor of left ventricular mass reduction one year after AVR. Meanwhile, Fabiani et al. found that miR-133 is upregulated in patients with AS and reduced ejection fraction, which we did not expect. We have summarized all the presented research in our review findings regarding miRNAs in AS in Figure 2.

The subjectivity of investigated miRNA choice is an essential limitation of many discussed studies. In most cases, the first step was a microarray, which found differently expressed miRNAs. The validation phase using qRT-PCR included only a few miRNAs, often chosen based von the extent of change in expression level or by the ‘literature review’. Therefore, instead of validating results obtained in previous research with some potential new findings, almost every study focused on a different miRNA, which led to a multitude of data that should now be validated.

The molecular aspects of diseases are doubtlessly exciting and provide invaluable insights into their pathophysiology. Nevertheless, translating these findings, regarding biomarkers such as miRNAs, into clinical practice requires much effort, time, and further research with a focus on validating existing evidence.

## 4. Future Perspectives

Much research has already been conducted. Nevertheless, there is still much to be carried out. First, it is essential to validate obtained results in a larger cohort. Once the validation has been completed, the research should focus on creating a miRNA-based diagnostic panel for AS utilizing miRNAs with the most altered expression. In an ideal but not unrealistic scenario, such a panel could also help identify the severity of AS.

Secondly, miRNAs might help predict interventional treatment outcomes, either AVR or TAVI. In this context, they would be instrumental, especially for patients at the borderline of qualification for AVR or TAVI.

Last but not least, miRNAs may be used as therapeutic targets or therapeutics. Nevertheless, before such a breakthrough is made, there are many steps to be taken. Crucially, the cause-and-effect relationship between miRNA expression and AS deterioration must be determined. Only then can preclinical studies, potentially followed by clinical ones, lead to novel, miRNA-based therapeutics used in AS treatment.

## Figures and Tables

**Figure 1 ijms-24-13095-f001:**
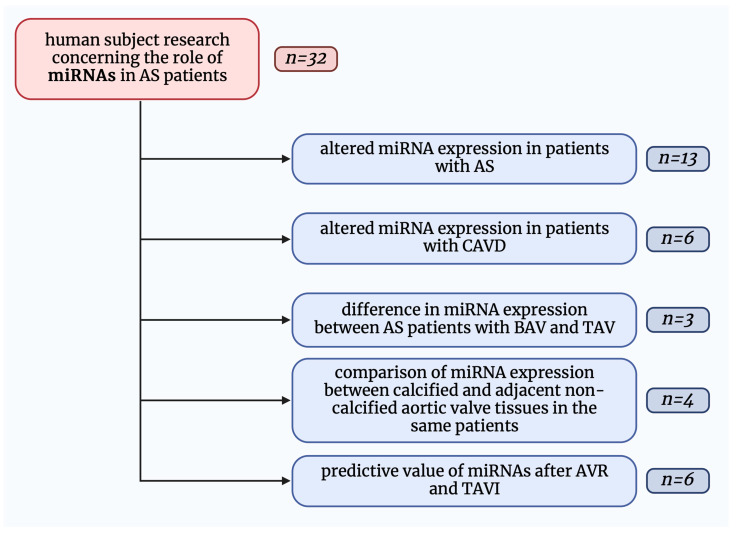
Graphical presentation of dividing studies included in this review. AS—aortic stenosis; AVR—aortic valve replacement; BAV—bicuspid aortic valve; CAVD—calcific aortic valve disease; miRNAs—micro-ribonucleic acids; *n*—a number of included studies in a given field; TAV—tricuspid aortic valve; TAVI—transcatheter aortic valve implantation.

**Figure 2 ijms-24-13095-f002:**
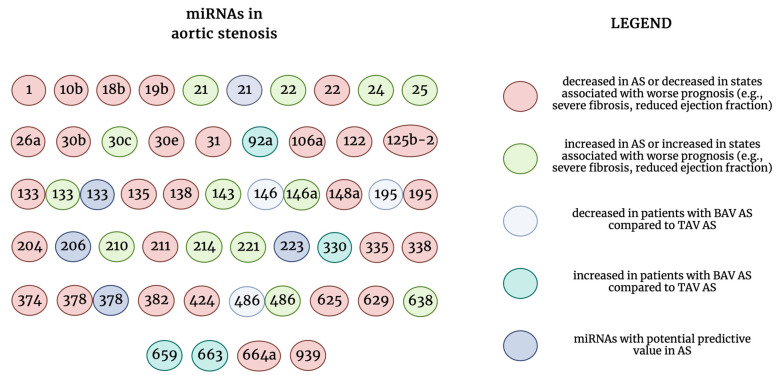
The summary of the role of microRNAs in aortic stenosis with division into several features. MicroRNAs with inconsistent results or more than one feature were presented more than once. AS—aortic stenosis; BAV—bicuspid aortic valve; miRNAs—micro-ribonucleic acids; TAV—tricuspid aortic valve.

**Table 1 ijms-24-13095-t001:** Summary of recent studies regarding altered microRNA expression in patients with aortic stenosis.

Ref.	Year	Population	Comparison	miRNA	Outcome	Methodology
[12]	2010	4 BAV AS pts undergoing AVR	5 BAV AI pts undergoing AVR	miR-26a miR-195	↓ miR-26a, miR-195 in AS pts	miRs in aortic valve samples via qRT-PCR
[13]	2013	75 AS pts undergoing AVR	32 surgical controls (myocardial expression) 25 HCs (plasma levels)	miR-21	↑ miR-21 in AS pts (both myocardial and plasma levels)	miR in left ventricle samples and in plasma via qRT-PCR
[14]	2014	28 AS pts undergoing AVR	pts divided by CVF: 15 SF pts, 13 non-SF pts	miR-18b miR-122	↓ miR-18b, miR-122 in SF pts compared to non-SF pts	miRs in myocardium samples via qRT-PCR
[15]	2014	57 AS pts	10 HCs	miR-210	↑ miR-210 in AS pts	miR in serum via qRT-PCR
[16]	2015	94 AS pts (with and w/o CAD)	101 controls (with and w/o CAD)	miR-21-5p miR-22-3p miR-24-3p miR-382-3p miR-451a	with CAD: ↑ miR-22-3p, miR-24-3p, and ↓ miR-382-3p in AS pts compared to controls w/o CAD: ↑ miR-21-5p, and ↓ miR-22-3p in AS pts compared to controls	miRs in plasma via qRT-PCR
[17]	2016	16 AS pts undergoing AVR	36 non-diseased cadavers	miR-21-5p miR-30e-5p miR-122-5p miR-221-3p miR-625-5p	↓ miR-30e-5p, miR-122-5p, miR-625-5p in AS pts ↑ miR-21-5p, miR-221-3p in AS pts	miRs in aortic valve samples via qRT-PCR
[18]	2016	28AS pts undergoing AVR subdivided: (i) 16 pts with low CMAI (group 1) (ii) 12 pts with high CMAI (group 2)	10 cardiovascular disease-free control subjects	miR-10b miR-125b-2* miR-338-3p	↓ miR-10b, miR-125b-2*, miR-338-3p in AS pts compared to controls ↓ miR-10b, miR-125b-2*, miR-338-3p in group 2 compared to group 1	miRs in myocardium via qRT-PCR
[19]	2016	46 AS pts (29 after AVR and 17 after TAVI)	46 controls	miR-30c miR-31 miR-106a miR-148a miR-204 miR-211 miR-424	↑ miR-30c and ↓ miR-31, miR-106a, miR-148a, miR-204, miR-211, miR-424 in AS pts	miRs in COPCs via qRT-PCR
[20]	2017	28 AS pts undergoing AVR	19 HCs for blood measurements necropsies in 10 cardiovascular disease-free control subjects	miR-19b miR-133a	↓ miR-19b, miR-133a in AS pts both in myocardium and blood samples	miRs in myocardium and blood via qRT-PCR
[21]	2017	8 AS pts undergoing AVR	8 cardiomyopathy pts undergoing heart transplantation	miR-204 miR-486	↓ miR-204 and ↑ miR-486 in AS pts	miRs in AVICs via qRT-PCR
[22]	2017	8 AS pts undergoing AVR	8 AI pts undergoing AVR	miR-26a miR-214 miR-374b* miR-939	↓ miR-26a, miR-374b*, miR-939 and ↑ miR-214 in AS pts	miRs in aortic valves samples via qRT-PCR
[23]	2019	16 AS pts undergoing AVR	3 HCs	miR-21 miR-24 miR-143	↑ miR-21, miR-24, miR-143 in AS pts miR-143 regulates MGP expression	miRs in aortic valve samples via qRT-PCR
[24]	2019	26 atherosclerotic AS pts undergoing AVR	32 non-atherosclerotic AS pts undergoing AVR	miR-146a	↑ miR-146a in atherosclerotic compared to non-atherosclerotic aortic valves	miR in aortic valve samples via qRT-PCR

↑—increased, ↓—decreased, AI—aortic insufficiency, AS—aortic stenosis, AVICs—aortic valve interstitial cells, AVR—aortic valve replacement, BAV—bicuspid aortic valve, CAD—coronary artery disease, CMAI—cardiomyocyte apoptotic index, COPCs—circulating osteogenic progenitor cells, CVF—collagen volume fraction, HCs—healthy controls, MGP—matrix gla protein, miR—microRNA, pts—patients, qRT-PCR—quantitative reverse transcription-polymerase chain reaction, ref.—reference, SF—severe fibrosis, TAVI—transcatheter aortic valve implantation.

**Table 2 ijms-24-13095-t002:** Summary of recent studies regarding altered microRNA expression in patients with calcific aortic valve disease.

Ref.	Year	Population	Comparison	miRNA	Outcome	Methodology
[25]	2019	8 CAVD pts undergoing AVR	8 cardiomyopathy pts undergoing heart transplantation	miR-204	↓ miR-204 in valves from CAVD pts	miR in aortic valve samples via qRT-PCR
[26]	2019	8 CAVD pts undergoing AVR	8 AI pts undergoing AVR	miR-214	↑ miR-214 in CAVD pts both in blood and aortic valve samples	miR in blood and aortic valve samples via qRT-PCR
[27]	2022	5 CAVD pts undergoing AVR	5 pts undergoing heart transplantation procedures	miR-135-5p miR-204-5p miR-335-3p miR-664a-3p	↓ miR-135-5p, miR-204-5p, miR-335-3p, and miR-664a-3p in valves from CAVD pts	miRs in aortic valve samples via qRT-PCR
[28]	2022	3 CAVD pts undergoing AVR	3 non-CAVD pts undergoing AVR	miR-629-3p	↓ miR-629-3p in valves from CAVD pts	miR in aortic valve samples via qRT-PCR
[29]	2022	10 CAVD pts undergoing AVR	5 aortic regurgitation pts (non-calcified aortic valves)	miR-138-5p	↓ miR-138-5p in valves from CAVD pts	miR in aortic valve samples via qRT-PCR
[30]	2022	33 CAVD pts undergoing AVR	12 pts undergoing heart transplantation	miR-22	↑ miR-22 in valves from CAVD pts	miR in aortic valve samples via qRT-PCR

↑—increased, ↓—decreased, AI—aortic insufficiency, AVR—aortic valve replacement, CAVD—calcific aortic valve disease, miR—microRNA, pts—patients, qRT-PCR—quantitative reverse transcription-polymerase chain reaction, ref.—reference.

**Table 3 ijms-24-13095-t003:** Summary of recent studies regarding the difference in microRNA expression between aortic stenosis patients with bicuspid and tricuspid aortic valves.

Ref.	Year	Population	Comparison	miRNA	Outcome	Methodology
[31]	2017	21 BAV AS pts undergoing AVR	29 TAV AS pts undergoing AVR	miR-195 miR-486	↓ miR-195, miR-486 in BAV AS pts	miRs in aortic valves samples via qRT-PCR
[32]	2017	17 BAV AS pts undergoing AVR	30 TAV AS pts undergoing AVR	miR-92a	↑ miR-92a in BAV AS pts	miRs in aortic valves samples via qRT-PCR
[33]	2020	15 BAV AS pts undergoing AVR	15 TAV AS pts undergoing AVR	miR-146 miR-330-3p miR-659 miR-663	↑ miR-330-3p, miR-659, miR-663 and ↓ miR-146 in BAV AS pts	miRs in aortic valve samples via qRT-PCR

↑—increased, ↓—decreased, AS—aortic stenosis, AVR—aortic valve replacement, BAV—bicuspid aortic valve, miR—microRNA, pts—patients, qRT-PCR—quantitative reverse transcription-polymerase chain reaction, ref.—reference, TAV—tricuspid aortic valve.

**Table 4 ijms-24-13095-t004:** Summary of recent studies regarding the comparison of microRNA expression between calcified and adjacent non-calcified aortic valve tissues in the same patients.

Ref.	Year	Population	Comparison	miRNA	Outcome	Methodology
[34]	2014	10 CAVD pts undergoing AVR (calcific aortic valves containing calcific nodules)	the same 10 CAVD pts undergoing AVR (adjacent non-calcific aortic valve tissues)	miR-30b	↓ miR-30b in calcific compared to non-calcific aortic valves	miR in aortic valve samples via qRT-PCR
[35]	2019	10 CAVD pts undergoing AVR (calcific aortic valves containing calcific nodules)	the same 10 CAVD pts undergoing AVR (adjacent non-calcific aortic valve tissues)	miR-638	↑ miR-638 in calcific compared to non-calcific aortic valves	miR in aortic valve samples via qRT-PCR
[36]	2019	10 DCAVD pts undergoing AVR (calcified valve)	the same 10 DCAVD pts undergoing AVR (non-calcified tissue)	miR-138	↓ miR-138 in calcific compared to non-calcific aortic valves	miR in aortic valve samples via qRT-PCR
[37]	2021	20 CAVD pts undergoing AVR (aortic valve tissues)	the same 20 CAVD pts undergoing AVR (adjacent normal tissues)	miR-25-3p	↑ miR-25-3p in aortic valve tissues	miR in tissues via qRT-PCR

↑—increased, ↓—decreased, AVR—aortic valve replacement, CAVD—calcific aortic valve disease, DCAVD—degenerative calcific aortic valve disease, miR—microRNA, pts—patients, qRT-PCR—quantitative reverse transcription-polymerase chain reaction, ref.—reference.

**Table 5 ijms-24-13095-t005:** Summary of recent studies regarding the predictive value of microRNAs after aortic valve replacement and transcatheter aortic valve implantation.

Ref.	Year	Population	Comparison	miRNA	Outcome	Methodology
[38]	2011	46 AS pts undergoing AVR	correlation btw miR levels and LVM reduction	miR-133a	↑ miR-133a an independent predictor of LVM reduction 1 year after AVR	miR in left ventricle samples via qRT-PCR
[39]	2013	74 AS pts undergoing AVR	correlation btw miR levels and LVM reduction	miR-133a	↑ miR-133a an independent predictor of LVM reduction 1 year after AVR	miR in plasma via qRT-PCR
[40]	2014	112 AS pts	40 HCs	miR-1 miR-133 miR-378	↓ miR-1, miR-133, miR-378 in AS pts ↓ miR-378 an independent predictor for LVH in AS pts	miRs in plasma via qRT-PCR
[41]	2017	30 AS pts undergoing TAVI	predictive value of miRNAs for cardiac function after TAVI	miR-206	miR-206 negatively correlated with LVEF	miRs in serum via qRT-PCR
[42]	2018	80 severe AS pts	pts divided by TTE parameters	miR-1 miR-21 miR-29 miR-133	↑ miR-1, miR-133 in pts with reduced EF ↑ miR-1, miR-21 in pts with LF condition ↑ miR-21 in pts with reduced global longitudinal strain at 1 year FU predictors: miR-21 reverse remodeling, miR-29 systolic function increase	miRs in blood via qRT-PCR
[43]	2022	65 severe AS pts undergoing TAVI	miRNA levels before and after TAVI predictive value of miRNAs for MACCE	miR-125a-5p miR-125b miR-223	↑ miR-125b and miR-223 after TAVI compared to before TAVI ↓ miR-223 before TAVI a predictor of MACCE (only in univariate analysis)	miRs in platelet-depleted plasma via qRT-PCR

↑—increased, ↓—decreased, AS—aortic stenosis, AVR—aortic valve replacement, btw—between, FU—follow-up, LF—low flow, LVEF—left ventricular ejection fraction, LVH—left ventricle hypertrophy, LVM—left ventricular mass, MACCE—major adverse cardiac and cerebrovascular events, miR—microRNA, qRT-PCR—quantitative reverse transcription-polymerase chain reaction, ref.—reference, TAVI—transcatheter aortic valve implantation, TTE—transthoracic echocardiography.

## Data Availability

Not applicable.

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
