# Peer review of "The Role of MicroRNAs in Aortic Stenosis—Lessons from Recent Clinical Research Studies"

_ijms, 2023, doi:10.3390/ijms241713095_

Round 1

Reviewer 1 Report

This review comprehensively summarized the role of microRNAs in Aortic Stenosis from Human Subject Research. The structure of the review is well-organized, and its content provides some informative insights. However, it appears that the authors have not delved into the literature in great depth and have instead presented a superficial overview that appears to be a compilation of statements from various papers. My specific comments are as follows:

1. The authors could go beyond compiling statements from various papers and instead engage in a more critical analysis that involves comparing differing viewpoints within the literature. By identifying prevailing trends, highlighting points of contention, and exploring emerging hypotheses, the review could evolve into a more intellectually stimulating resource.

2. Future direction part is needed. The review should not only summarize the current state of knowledge but also stimulate new avenues for research and clinical applications.

Reviewer 2 Report

The authors provide a thorough review of studies focused on miRNAs in aortic stenosis and their potential utilization as indicators for treatment of this condition.  The review focuses primarily on human studies.  In general, the manuscript is well-written and will be of interest to individuals in this field.  Several relatively minor suggestions:

- In several places, the authors indicate "roles" for particular miRNAs; however, some of the studies appear to only evaluate expression of miRNAs.  I would limit the use of "roles" to data where functional studies were actually performed.  

- The tables are very helpful and effectively summarize a lot of data.  I suggest referring to the tables earlier in the respective section of the manuscript.

- The authors should consider expanding the Conclusions section to include potential next directions in this field.  What will be required before miRNAs can be utilized as biomarkers or therapeutic targets.

- I would also include limitations of some of the studies discussed in the Conclusion section.  This includes the lack of temporal analyses in many cases.

The manuscript is well-written and needs minor grammatical editing.

Reviewer 3 Report

MicroRNAs (miRNAs) are small non-coding RNA molecules that play a crucial role in post-transcriptional regulation of gene expression. They function by binding to messenger RNA (mRNA) molecules and thereby influencing the translation of those mRNAs into proteins. MiRNAs have been implicated in various biological processes, including development, cellular differentiation, proliferation, and disease. MiRNAs have been found to regulate several processes relevant to aortic stenosis: valvular calcifications, inflammation, extracellular matrix remodeling, cellular senescence, apoptosis, and smooth muscle cell proliferation.

I only had one comment. This work is excellent, but may provide the information about mechanism of AS and add some paragraphs about the mechanism and microRNA involvement.
